# ERα-Dependent Regulation of Adropin Predicts Sex Differences in Liver Homeostasis during High-Fat Diet

**DOI:** 10.3390/nu14163262

**Published:** 2022-08-10

**Authors:** Clara Meda, Arianna Dolce, Elisabetta Vegeto, Adriana Maggi, Sara Della Torre

**Affiliations:** 1Department of Health Sciences, University of Milan, 20146 Milan, Italy; 2Department of Pharmaceutical Sciences, University of Milan, 20133 Milan, Italy

**Keywords:** liver, metabolism, NAFLD, high-fat diet, dietary lipids, hepatokines, adropin, sex and gender differences, estrogens, estrogen receptor alpha

## Abstract

Non-alcoholic fatty liver disease (NAFLD) represents a public health issue, due to its prevalence and association with other cardiometabolic diseases. Growing evidence suggests that NAFLD alters the production of hepatokines, which, in turn, influence several metabolic processes. Despite accumulating evidence on the major role of estrogen signaling in the sexually dimorphic nature of NAFLD, dependency of hepatokine expression on sex and estrogens has been poorly investigated. Through in vitro and in vivo analysis, we determined the extent to which hepatokines, known to be altered in NAFLD, can be regulated, in a sex-specific fashion, under different hormonal and nutritional conditions. Our study identified four hepatokines that better recapitulate sex and estrogen dependency. Among them, adropin resulted as one that displays a sex-specific and estrogen receptor alpha (ERα)-dependent regulation in the liver of mice under an excess of dietary lipids (high-fat diet, HFD). Under HFD conditions, the hepatic induction of adropin negatively correlates with the expression of lipogenic genes and with fatty liver in female mice, an effect that depends upon hepatic ERα. Our findings support the idea that ERα-mediated induction of adropin might represent a potential approach to limit or prevent NAFLD.

## 1. Introduction

Non-alcoholic fatty liver disease (NAFLD) is a spectrum of liver diseases, ranging from hepatic steatosis to fibrosing non-alcoholic steatohepatitis (NASH), cirrhosis, and hepatocellular carcinoma (HCC) [1]. Despite the growing public health impact of NAFLD [2], treatment options remain limited, probably as a consequence of the limited understanding of the biological drivers of the heterogeneity of NAFLD pathogenesis and progression [3,4].

NAFLD incidence is lower in fertile women compared to men and post-menopausal women [5,6], suggesting a pivotal role exerted by the estrogen signaling in counteracting NAFLD development and progression [7,8]. In the liver of fertile female mice, estrogens act mainly through estrogen receptor alpha (ERα), which modulates hepatic metabolism according to the energy requirements peculiar to each reproductive stage [9,10,11]. Possibly as a consequence both of the strict interplay between metabolism and reproduction that the female liver acquired during evolution [12] and of the sex-dimorphic expression of hepatic ERα [9,10], the regulation of lipid metabolism is significantly different in the liver of the two sexes, contributing to sex differences in NAFLD susceptibility. Indeed, when exposed to an excess of dietary lipids, female mice, contrary to males, limit liver lipid deposition, an ability that depends on the hepatic ERα [13]. Further supporting a key role for hepatic ERα signaling in counteracting NAFLD progression, ERα is poorly expressed in the liver of NASH compared to NAFLD patients [14].

Growing evidence suggests that NAFLD may precede dysfunctions in other organs during the pathogenesis of systemic metabolic diseases, making NAFLD a risk factor for other cardiometabolic diseases, including diabetes and atherosclerosis [15,16,17,18,19]. In addition to other signaling pathways, NAFLD alters the production of many hepatokines [20,21,22,23], key messengers that, once secreted in response to nutrients and other factors, reach other organs, where they influence several metabolic processes. Altered hepatokine secretion at NAFLD onset can significantly impair the inter-organ signaling crosstalk, which, in turn, can trigger the progression of complex and multifaceted metabolic dysregulation, leading to systemic metabolic diseases. 

Recent advances indicated that some hepatokines, including fetuin-A and fetuin-B [24], hepassocin [25], leucocyte cell-derived chemotaxin 2 (LECT2) [26], follistatin [27], retinol-binding protein 4 (RBP4) [28], selenoprotein P (SeP/Sepp1) [29], and tsukushi (Tsku) [30] promote NAFLD progression, while others, including fibroblast growth factor 21 (FGF21) [31], adropin [32], and growth differentiation factor 15 (GDF15) [33], limit this process [21,22,23]. The dependency of hepatokine expression on sex and estrogens has been scarcely investigated even though the role of estrogen in the sexually dimorphic nature of NAFLD is known [7].

In the present study, we evaluated sex-specific changes in the hepatic expression of the most relevant hepatokines known to be altered in NAFLD. Given the role of ERα signaling in hepatic sexual dimorphism and in metabolic homeostasis [9,10,13,34], we performed our analysis by comparing control (syngenic, ERα floxed) and liver-specific ERα KO (LERKO) mice of both sexes. Our data identify adropin as the most predictive hepatokine, featuring a sex-specific ability to preserve liver homeostasis under conditions of dietary lipid excess.

## 2. Materials and Methods

### 2.1. Animals

In the present study, we used control (ERα floxed), LERKO (liver-specific ERα knockout), and ERE-Luc mice, all C57BL/6J strain. Mice were fed *ad libitum* with a standard diet (4RF21 standard diet, Mucedola, Settimo Milanese, Italy) and provided with filtered water. The animal room was maintained within a temperature range of 22–25 °C, relative humidity of 50 ± 10%, and under an automatic cycle of 12 h light, 12 h dark (lights on at 07:00 a.m.). 

For the sex differences experiment, intact female mice were euthanized when in the *Proestrus* or *Metestrus* phase of the estrous cycle, characterized by high and low levels of estrogens, respectively, with vaginal smears done at 9:00 a.m. 

For OVX experiments, female mice of 2 months of age were anesthetized with an s.c. injection of 70 μL of ketamine (109.2 mg/kg Ketavet 100, Intervet, Aprilia, Italy) and xylazine (8.4 mg/kg Rompun, Bayer, Milan, Italy) solution and then ovariectomized (OVX) or sham (SHAM) operated.

For the HRT experiment, three weeks after surgery, mice were assigned to a specific experimental group and treated *per os* (gavage) with vehicle (2% Tween-80 and 0.5% carboxymethylcellulose water solution), 3 mg/kg conjugated estrogen (CE), 10 mg/kg bazedoxifen (BZA), 10 mg/kg BZA in association with 3 mg/kg CE (TSEC), or 10 mg/kg raloxifen (RAL) for 21 days.

For the HFD experiment, control (ERα floxed) and LERKO (liver ERα knockout) mice were fed with a control, low-fat diet (Research Diet, D12450B) or with a high-fat diet (HFD; Research Diets, D12492) for 16 weeks starting at 4 months of age.

SHAM female mice were collected when in the *Metestrus* phase of the estrous cycle with vaginal smears done at 9:00 a.m.

In all experiments, to avoid any possible confounding effect due to the circadian rhythm or feeding status [35], mice were euthanized in the early afternoon after 6 h of fasting.

### 2.2. Cell Culture and Treatments

HepG2 cells were obtained from American Type Culture Collection—LGC Standards. Cells were maintained in DMEM high glucose (Life Technologies) with 10% FBS, supplemented with 1% mixture of antibiotics containing penicillin, streptomycin, and amphotericin B (Life Technologies, Monza, Italy). HepG2 cells were seeded at a density of 0.55 × 10^6^ cells/well in 6-well plates and incubated at 37 °C in the presence of 5% CO_2_ for 24 h. The medium was then changed and replaced with DMEM without phenol red (Life Technologies) with 10% DCC-FBS, supplemented with 1% mixture of antibiotics containing penicillin, streptomycin, and amphotericin B (Life Technologies). Before treatment, medium was replaced with DMEM without phenol red with 1% DCC-FBS, supplemented with 1% mixture of antibiotics containing penicillin, streptomycin, and amphotericin B. Cells were treated with vehicle (ethanol) or 1 nM 17β-estradiol (E2) (Merck-Sigma) for 16 and 24 h.

Control and LERKO mice were given 50 μg/Kg 17β-estradiol (E2) or vehicle (VEH) by subcutaneous injection; 16 h after, adult hepatocytes were isolated and grown according to the protocol followed by Valverde et al. [36]. Prior to treatment, cells were incubated in serum and phenol red-free medium for 45 min at 37 °C; then cells were exposed to VEH or 1 nM E2, accordingly with the previous treatment in vivo, for 2 h, before gene expression analysis.

### 2.3. RNA-Sequencing and Transcriptomics Data Analysis

RNA-sequencing and transcriptomics data analysis were performed as previously described in [9,34]. 

### 2.4. Reverse Transcription Quantitative Real-Time PCR (RT-PCR) Gene Expression Analysis

Total liver RNA was isolated with TRIzol Reagent (Invitrogen, Monza, Italy) and purified using the RNeasy mini-kit protocol (Qiagen, Milano, Italy), according to the manufacturer’s instructions. The preparation of cDNA was done by using random primers (Promega, Milano, Italy), deoxynucleotide triphosphate (GE Healthcare, Casoria, Italy) and Moloney murine leukemia virus reverse transcriptase (M-MLV RT) (Promega), according to the manufacturer’s instructions. For RT-PCR experiments, the reaction mix for each sample was made up of 2 μL of pre-diluted cDNA, 5 μL of 2X Master Mix (Luna^®^ Universal Probe qPCR Master Mix, BM3004L or Promega, A6001), 0.2 μL of 10 mM primers or 0.5 μL of 20x TaqMan Gene Expression Assays (ThermoFisher, Monza, Italy), in a final volume of 10 μL. The primers used for RT-PCR reactions are listed in Appendix A. *Rplp0* and *36b4* were used as reference gene assays for mouse and human samples, respectively. The reaction was conducted according to the manufacturer’s protocol using a QuantStudio™ 3 Real-Time PCR System with the following thermal profile: 2 min at 95 °C; 40 cycles with 15 s 95 °C, 1 min at T_annealing_ (for Sybr Green chemistry) and 1 min 95 °C; 40 cycles with 15 s 95 °C, 1 min at 60 °C (for TaqMan chemistry). The data were analyzed using the 2^−∆∆Ct^ method [37].

### 2.5. Biochemical Assays

Triglyceride levels were measured with appropriate kits according to the manufacturer’s protocols (Biovision and Vinci-Biochem, Vinci, Italy).

### 2.6. Quantification, Statistical Analysis and Figure Preparation

Correlation scatter plots were performed by using Origin (Pro) (Adalta, Arezzo, Italy). All statistical analyses were done using GraphPad Prism 8.0 (GraphPad Software, San Diego, CA, USA). Multiple testing comparisons were done by one or two-way ANOVA followed by Bonferroni’s *post hoc* test; two-tailed Student’s *t*-test was used for comparisons between two experimental groups. All data are expressed as mean ± SEM. A *p* value less than 0.05 was considered statistically significant. All statistical parameters are expressed in the figure legends. Metabolomic data analysis was performed by using Metaboanalyst 5.0 software (http://www.metaboanalyst.ca/ (accessed on 9 July 2022)). 

Figures are adapted from images created with BioRender https://biorender.com/ (accessed on 11 July 2022)).

## 3. Results

### 3.1. Liver mRNA Content of Hepatokines Depends on Sex and Estrogen Levels

Starting from the idea that estrogen-dependent control of hepatic sexual dimorphism may account for the sex-specific susceptibility to NAFLD [38,39], we hypothesized that the transcriptional regulation of hepatokines differs between males and females, even under physiological conditions. To test this hypothesis, we enquired available RNA-Seq dataset (Bioproject PRJNA395963; [9]) to evaluate the gene expression of the most relevant hepatokines known to be altered in NAFLD [21,22,23] by comparing male (MAL) and female (FEM) mice in two phases of the estrous cycle characterized by low and high levels of estrogens (E2), respectively.

As shown in Figure 1A, several hepatokines are differentially expressed in the liver of the two sexes. In particular, mRNA levels of *Angptl3* (angiopoietin like 3; *p* = 0.001), *Dpp4* (dipeptidyl peptidase 4; *p* = 0.0003), *Fetuin A* (*p* = 0.007), *Igfbp1* (insulin-like growth factor binding protein 1; *p* = 0.027), *Lect2* (*p* = 0.01), *Postn* (periostin; *p* = 0.004), *Sepp1* (*p* = 0.005), *Serpinb1a* (serpin family B member 1a; *p* = 0.033), *Serpinf1* (serpin family F member 1; *p* = 0.04), *Smoc1* (SPARC-related modular calcium binding 1; *p* = 0.001) were higher, while mRNA levels of adropin (which is encoded by *Enho*, energy homeostasis associated; *p* = 0.013) and *Inhbe* (inhibin subunit beta E; *p* = 0.014) were reduced in the liver of females with low E2 in comparison to males (Figure 1A).

When we compared males to females with high E2 levels, sex differences in liver mRNA content of *Angptl3*, *Dpp4*, *Fetuin A*, and *Lect2* were mostly retained, while others were boosted (*Serpinb1a*), newly acquired (*Angptl4*, angiopoietin like 4, *p* = 0.003; *Igf1*, insulin like growth factor 1, *p* = 0.011; *Igfbp3*, insulin-like growth factor binding protein 3, *p* = 0.014), or lost (*Enho*; *Igfbp1*; *Inhbe*; *Postn*; *Sepp1*; *Serpinf1*; *Smoc1*) with respect to the comparison of males/females with low E2 (Figure 1A). Of relevance, *Angptl3* (*p* = 0.001), *Angptl4* (*p* = 0.049), *Angptl6* (angiopoietin like 6; *p* = 0.013), *Sepp1* (*p* = 0.042), *Smoc1* (*p* = 0.0003), and *Tsku* (*p* = 0.019) were significantly modulated by estrogens in the liver of fertile females (Figure 1A).

Since the lack of estrogens at menopause increases the risk of developing NAFLD and associated co-morbidities, also through the alteration of hepatokines, we next examined the hepatic mRNA content of hepatokines in a mouse model of menopause (ovariectomy, OVX) by enquiring available RNA-Seq data (Bioproject PRJNA778593; [34]). The lack of estrogens significantly induced the mRNA levels of *Angptl4* (*p* = 0.019), *Rbp4* (*p* = 0.014), and *Tsku* (*p* = 0.017), while it reduced the mRNA content of *Enho* (*p* = 0.018), *Fetuin A* (*p* = 0.005), *Sepp1* (*p* = 0.047), and *Serpinf1* (*p* = 0.047) in the liver of females 4 months after OVX (Figure 1B).

To identify the hepatokines that better recapitulate sex- and estrogen-dependent effects, we plotted the relative gene expression of the hepatokines measured in fertile females over males (FEM/MAL) and in OVX over SHAM (OVX/SHAM). By considering a Log_2_-fold change (FC) > 1 or < −1, we selected four hepatokines, namely *Enho*, *Inhbe*, *Igfbp1*, and *Tsku* (Figure 1C). 

In order to further explore the dependency of the four hepatokines on the estrogen signaling, we investigated the presence of classical estrogen-responsive elements (EREs) in their promoters (Figure 1D) by the means of EPD and JASPAR databases. By considering a relative profile score threshold > 80%, we found at least one ERE in the region −/+1000 around the TSS (transcription start site) of all the four hepatokines (Figure 1E), suggesting that estrogen-dependent regulation of these hepatokines requires hepatic ERα.

### 3.2. Estrogen Treatment Regulates Hepatokine mRNA Content in Hepatic Cells

To determine whether the identified hepatokines are directly regulated by estrogens, we conducted in vitro studies in HepG2 cells treated with 1 nM 17β-estradiol (E2) (Figure 2A).

In response to E2 administration, ENHO mRNA levels did not change, while INHBE was not detectable in our experimental conditions (Figure 2B). Conversely, IGFBP1 mRNA levels increased by 96% 16 h after treatment, a trend lost at 24 h (Figure 2B). Alternatively, the hepatic mRNA content of TSKU showed a time dependent bi-phasic regulation by E2, being increased by 32% and decreased by 26% 16 h and 24 h after treatment, respectively (Figure 2B).

To further explore the relevance of hepatic ERα in the direct regulation of the four hepatokines by E2, we analyzed its mRNA levels in primary hepatocyte cells derived from male and female control and LERKO mice treated with VEH or E2 (Figure 2C; see also Methods section). As expected, *Esr1* mRNA levels were higher in hepatocytes from control females compared to those derived from control males and not detectable in hepatocytes from LERKO mice of both sexes (Figure 2D).

E2 treatment significantly induced the mRNA levels of *Enho* only in hepatocytes derived from control males, leaving unaffected the *Enho* mRNA content in female control hepatocytes. On the contrary E2 treatment did not modify *Enho* mRNA in the hepatocytes of LERKO male mice, while we observed a trend to decrease, still not significant, in the cells derived from LERKO female mice (Figure 2E).

The Inhbe mRNA levels in the different groups suggest a sexually dimorphic expression that seems to depend upon ERα presence. Indeed, Inhbe mRNA levels were slightly higher (still without reaching significance) in hepatocytes from control females compared to males, while the lack of ERα led to significantly lower levels of Inhbe mRNA in hepatocytes from LERKO females compared to those derived from LERKO males (Figure 2F). Treatment with E2 enhanced Inhbe mRNA content in hepatocytes from both sexes in controls, especially in those derived from females (Figure 2F). By contrast, in the hepatic cells from LERKO mice, E2 treatment gave opposite results: the mRNA content of Inhbe was strongly inhibited by E2 in LERKO male hepatocytes and unaffected in LERKO female cells (Figure 2F). In contrast, Igfbp1 mRNA levels did not change in our experimental conditions (Figure 2G).

Like Inhbe, Tsku also showed a sexually dimorphic expression that depends on ERα presence. In fact, Tsku mRNA levels were lower, still without reaching significance, in hepatocytes from control females compared to males, while, without ERα, vehicle-treated LERKO females expressed significantly more Tsku mRNA compared to male counterparts (Figure 2H). Estrogen treatment did not affect Tsku mRNA content in hepatocytes from control males, while slightly reduced it in those from control females, without reaching statistical significance (Figure 2H). Conversely, in hepatic cells from LERKO mice, E2 treatment induced Tsku mRNA levels in male-derived hepatocytes while it did not modify it in female ones (Figure 2H). Notably, with respect to control hepatocytes, the lack of hepatic ERα raised Tsku mRNA levels to higher values in hepatocytes from LERKO males when treated with E2 and in hepatocytes from LERKO females independently of the treatment (Figure 2H).

All the data obtained from in vitro studies support the idea that hepatic ERα mediates estrogen effects and regulates the expression of the four hepatokines focused on in this study, especially *Enho*, *Inhbe*, and *Tsku*, thus contributing to their sexually dimorphic expression.

### 3.3. Hormone Replacement Therapy Can Restore Hepatokine Expression in a Mouse Model of Menopause

Although debated [40,41,42], hormone replacement therapy (HRT) still represents a valuable approach for post-menopausal women to prevent or limit metabolic diseases that occur as a consequence of the lack of the regulatory activity of estrogens [43,44]. With the aim to clarify the potential of various HRTs in modulating the expression of this subset of hepatokines, we performed a short-term treatment (21 days) with conjugated estrogen (CE), bazedoxifen (BZA), BZA in association with CE (TSEC), or raloxifen (RAL) on OVX female mice. Then we analyzed liver gene expression by comparing HRT-treated OVX versus fertile (CYC) and vehicle (VEH)-treated OVX as reference controls (Figure 3A). 

As shown in Figure 3B–E, short-term OVX significantly reduced the liver mRNA content of *Enho* and *Igfbp1*, while it enhanced *Inhbe* mRNA levels. Differently from what is shown in Figure 1B, *Tsku* mRNA content was unaffected after short-term OVX, suggesting that the length of OVX might lead to progressive metabolic degeneration and differently impact on hepatokine regulation. 

CE restored mRNA contents of *Enho*, *Inhbe*, *Igfbp1*, and *Tsku* in OVX mice to the values measured in the liver of fertile females. Also BZA brought *Igfbp1* mRNA levels back to those of fertile females (Figure 3D), but it did not modify *Enho* and *Inhbe* mRNA levels, when compared to VEH-treated OVX mice (Figure 3B,C). As to *Tsku*, its levels were significantly induced by BZA in the liver of OVX females (Figure 3E). TSEC treatment left *Enho* mRNA contents unaffected, but decreased *Inhbe* mRNA levels to those of fertile females and induced the expression of *Igfbp1* mRNA content in the liver of OVX females, notably to levels even higher than those of cycling females (Figure 3B–D). Despite the unchanged mRNA values of *Enho*, TSEC treatment prevented the increase of *Inhbe* mRNA content and promoted the expression of *Igfbp1* in the liver of OVX females (Figure 3B–D). Of relevance, TSEC brought *Igfbp1* to levels higher than in fertile females, possibly as an additive effect mediated by the combined treatment with CE and BZA (Figure 3D). 

RAL was unable to bring *Enho, Inhbe*, and *Igfbp1* mRNA values back to those of fertile females, even if there is a trend, still not significant, to reduce *Inhbe* mRNA levels compared to OVX females (Figure 3B–D). RAL raised *Tsku* mRNA levels three times higher than those measured in VEH-treated fertile and OVX females (Figure 3E).

All together, these findings identify CE as the HRT that better restores mRNA content of *Enho*, *Inhbe*, *Igfbp1*, and *Tsku* in a mouse model of short-term menopause. 

### 3.4. Adropin mRNA Levels Negatively Correlate with Fatty Liver in a Mouse Model of NAFLD

Given the lower susceptibility of females to NAFLD and associated pathologies, we then investigated the extent to which the identified subset of hepatokines was modulated in the liver of male and female mice under high-fat diet (HFD). To explore the relevance of hepatic estrogen signaling, we included LERKO (liver-specific ERα KO) mice in our analysis (Figure 4A).

With respect to control diet, HFD enhanced *Enho* mRNA content in the liver of females (+270%), while slightly reduced it in males (−40%). This sex-specific modulation is partly mediated by hepatic ERα, since *Enho* was induced to a greater extent in LERKO males (+276%) but limited in LERKO females (−31%) with respect to control mice (Figure 4B).

HFD induced *Inhbe* mRNA content in both sexes (+63% and +75% for control males and females, respectively) by preserving a higher expression in males (+50%) than females (Figure 4C) in control-fed mice. In the absence of hepatic ERα, *Inhbe* showed an opposite regulation between the two sexes (+125% in LERKO females), which was unchanged by HFD (Figure 4C).

Hepatic mRNA levels of *Igfbp1* were higher in control females (+258%) than males when fed with control diet; under HFD, *Igfbp1* decreased only in females (−54%), keeping to higher (+83%) levels than in males (Figure 4D). Sex-specific modulation of *Igfbp1* mRNA depends on hepatic ERα, since *Igfbp1* levels did not differ in LERKO males and females (Figure 4D). Notably, just the loss of hepatic ERα significantly up-regulated *Igfbp1* mRNA contents in control-fed LERKO males, while HFD decreased *Igfbp1* mRNA levels, especially in LERKO females.

In mice fed with control diet, liver mRNA content of *Tsku* was sexually dimorphic, with females showing higher values (+208%) than males; HFD reverted this sex difference by increasing (+74%) and decreasing (−74%) *Tsku* mRNA in males and females, respectively (Figure 3E). This effect seems mostly independent of hepatic ERα, since it was similar in control and LERKO mice (Figure 4E).

In order to investigate the extent to which changes in hepatokine mRNA levels may represent a reliable marker of liver lipid deposition, we performed Pearson correlation analysis between the relative mRNA expression of the four hepatokines and the content of triglycerides (TG) in the liver. As shown in Figure 4F, in control mice of both sexes, *Enho* mRNA levels correlated with liver lipid deposition in a sex-specific fashion. In mice lacking hepatic ERα, only *Enho* and *Igfbp1* mRNA levels displayed a correlation with liver TG that was limited to LERKO males. Of relevance, linear regression between *Enho* mRNA and TG in LERKO males looks like the one observed in control females.

These results point to a sex and ERα-dependent regulation of *Enho*, *Inhbe*, and *Igfbp1* mRNA levels under HFD, and identify adropin as the hepatokine better reporting a sex-specific and ERα-dependent ability to counteract liver lipid deposition under an excess of dietary lipids.

### 3.5. Sex-Specific Reprogramming of Liver Metabolism in a HFD-Induced Model of NAFLD Depends on Hepatic ERα

Besides in skeletal muscles where it impacts fuel selection [45], adropin also plays important roles in the regulation of energy metabolism and metabolic homeostasis in the liver [46,47,48]. For this reason, we then focused our analysis on key metabolic genes known to be regulated by adropin [46,49,50,51,52].

Under the experimental conditions adopted, males and females did not differ in the hepatic mRNA content of Gck (glucokinase) when fed with control diet; HFD strongly induced Gck in males (+232%) and, to a lower extent, in females (+171%), leading to significantly higher mRNA levels of Gck in males (+48% vs. females) (Figure 5A). The lack of hepatic ERα limited the HFD-dependent induction of Gck in LERKO males (+91%), while it did not impair Gck modulation in LERKO females (Figure 5A).

Under standard diet, the mRNA content of Pklr (pyruvate kinase)—that catalyzes a rate-limiting step of glycolysis by converting phosphoenolpyruvate into pyruvate—is greater in the liver of females (+85%) compared to control males, a regulation boosted by hepatic ERα since LERKO females showed decreased mRNA levels of Pklr (−29% vs. control females) (Figure 5A). HFD regimen nullified this sex difference by keeping Pklr mRNA levels low in males and significantly reduced in females (−61%), respectively (Figure 5A). 

Given the relevance of adropin in limiting hepatic glucose production [52], we measured the mRNA content of key genes involved in gluconeogenesis, such as G6pc (glucose-6-phosphatase) and Pck1 (phosphoenolpyruvate carboxykinase 1). When fed with control diet, females expressed higher mRNA levels of G6pc (+83% vs. control males); the lack of hepatic ERα led to enhanced expression of G6pc in LERKO males (+52% vs. control males), thus abolishing sex differences (Figure 5B). HFD led to a slight increase in G6pc in males and to a significant reduction (−39%) in females, while no changes were observed in LERKO mice, pointing to the need of hepatic ERα for G6pc regulation under dietary excess of lipids (Figure 5B). Conversely, HFD triggered a similar reduction of Pck1 mRNA in both sexes, regardless of the genotype (Figure 5B), suggesting that hepatic ERα can be dispensable for Pck1 regulation under the experimental conditions evaluated. 

Adropin has been found to downregulate Pdk4 (pyruvate dehydrogenase kinase 4) mRNA content and increase glucose utilization in muscle cells [45]. Under physiological conditions, Pdk4 mRNA levels were higher in the liver of females (+275%) than males, independently of hepatic ERα. With HFD, hepatic Pdk4 mRNA levels increased in control males (+110%) but not LERKO males, while they did not change in females of either genotype (Figure 5C).

Since low levels of adropin worsen HFD-induced metabolic defects [53], probably by enhancing the de novo synthesis of lipids (DNL) [50], we analyzed the mRNA content of genes relevant for this pathway, such as Acaca (acetyl-CoA carboxylase alpha), Fasn (fatty acid synthase), Elovl6 (ELOVL fatty acid elongase), Lpin1 (lipin1), and Dgat2 (diacylglycerol O-acyltransferase 2) (Figure 5D,E). When fed with control diet, the mRNA content of Acaca (+383%), Fasn (+423%), Elovl6 (+481%), Lpin1 (+112%), and Dgat2 (+29%) was higher in females with respect to control males. Sex differences in DNL genes were partially dependent on hepatic ERα, since their mRNA levels were mostly limited in LERKO females with respect to control females (−53% for Acaca; −66% for Fasn; −48% for Elovl6). Under an excess of dietary lipids, control males promoted (+110% Fasn, +42% Elovl6 versus control diet) while females reduced (−45% Fasn, −68% Acaca, −74% Elovl6, −73% Lpin1 versus control diet) fatty acid synthesis and elongation. In mice lacking hepatic ERα, HFD gave sex-specific opposite trends; indeed, DNL genes were unaffected or reduced by HFD in LERKO males (+8% for Fasn, −26% for Acaca, −37% for Elovl6 versus CTRL diet); conversely, with the exception of Lpin1 (−73%), the magnitude of changes in the expression of these key genes was greatly blunted (−41% for Acaca and −60% for Elovl6) or even reverted (+51% for Fasn) in HFD-fed LERKO females with respect to control females (Figure 5D,E). 

Given the role of adropin in the regulation of substrate oxidation preferences in muscle [45], we analyzed the mRNA content of *Cpt1α* (carnitine palmitoyltransferase 1A), a gene codifying for a limiting enzyme involved in fatty acid β-oxidation (FAO). With respect to males, *Cpt1α* was more expressed in the liver of females for both genotypes (+22% and +23% in control and LERKO mice, respectively) when fed with control diet. Under HFD, *Cpt1α* mRNA levels increased in control (+48%) but not in LERKO males, while its content was not modified in female mice (Figure 5F). 

Lower adropin expression is associated with oxidative stress and NAFLD severity [32]; the antioxidant properties of adropin in NASH are mediated by the activation of NRF2 (nuclear factor erythroid 2–related factor 2) signaling, leading to liver protection against injury [54]. Under control diet, females showed a greater mRNA content of Nrf2 (+87%) with respect to control males; notably, the lack of hepatic ERα partially blunted this sex difference in LERKO mice (+35% LERKO females vs. LERKO males) (Figure 5G). HFD enhanced Nrf2 mRNA to the levels of females in control males but not LERKO males (Figure 5G); conversely, Nrf2 showed a trend to increase in control females (+13%) and a significant induction by HFD in LERKO females (+25%), suggesting that Nrf2 hepatic expression in a setting of excess dietary lipids can be modulated by hepatic estrogen signaling in a sex-specific fashion.

To deepen our analysis of NRF2 signaling, we evaluated the mRNA content of known gene targets, such as Hmox1 (heme oxygenase 1) and Nqo1 (NAD(P)H quinone dehydrogenase 1). With respect to control males, females had higher Hmox1 mRNA levels (+87%); HFD slightly increased Hmox1 mRNA content in males (+28%), while it strongly enhanced (+54%) its expression in females. With both diets, the lack of hepatic ERα did not affect Hmox1 mRNA levels in LERKO mice of both sexes, since LERKO exhibited similar profiles of Hmox1 as control mice (Figure 5G). In mice fed with control diet, hepatic ERα (known to be much more expressed in the liver of females) contributed to the sexually dimorphic expression of Nqo1 (+289% in females compared to males), indeed, the sex difference was smoothed in LERKO mice (only +67% in females with respect to males); notably, the lack of hepatic ERα led to an opposite regulation of Nqo1 by HFD in male and female LERKO (Figure 5G). Indeed, HFD enhanced Nqo1 mRNA content in control males (+110%), while it decreased it in LERKO males (−49%). On the contrary, in both control and LERKO females, HFD promoted Nqo1 induction (+21% and +35%, respectively) (Figure 5G).

All together, the data point to ERα as an important contributor to the differential response of the two sexes to an excess of dietary lipids, highlighting its role in the sex difference observed in metabolic adaptation.

### 3.6. Adropin Induction Negatively Correlates with Lipid Synthesis in Female Mice

To evaluate the extent to which the modulation of adropin accounts for the sex-specific metabolic adaptation to HFD with limited lipid deposition in the liver, we performed a Pearson correlation analysis between the mRNA content of *Enho* and mRNA levels of all the measured genes known to be involved in the regulation of hepatic metabolism. As shown in Figure 6A,B, hepatic *Enho* mRNA levels negatively correlated with the hepatic content of *Acaca*, *Fasn*, *Elovl6*, *Dgat2*, and *Lpin1* in female but not male control mice, suggesting that the induction of *Enho* observed in the liver of control females can, at least in part, mediate the reduction in lipid synthesis, thus limiting fat deposition in the liver. The female-specific negative correlation between *Enho* and lipogenic genes is at least in part mediated by hepatic ERα, since it was lost or smoothed in LERKO mice (Figure 6C,D). Notably, *Enho* correlation with *Pklr* was positive in control males and negative in females of both genotypes, probably leading to opposite changes in mitochondrial pyruvate flux and incorporation into citrate, that might contribute to an opposite regulation of DNL in the two sexes (Figure 6).

We then evaluated the relevance of the observed changes in mRNA content on the regulation of hepatic metabolites measured in [13] by the means of Metaboanalyst software. As shown in Figure 7A, HFD differently affected the metabolite–metabolite interaction in the liver of male and female mice, also in dependency of hepatic ERα. In particular, with respect to males, HFD influenced a few metabolites in the liver of control females (Figure 7A). Conversely, HFD led to a great reprogramming of liver metabolites in LERKO males. In LERKO females, changes in metabolite-metabolite interaction included several more metabolites than in control females, supporting the idea that females better preserve liver homeostasis, an effect strongly dependent on hepatic ERα (Figure 7A). According to this, the integrative analysis of the transcriptomic and metabolomic data indicated that changes in key genes and metabolites greatly impacted on fatty acid biosynthesis only in the liver of control females, but not in control males or LERKO of both sexes which, indeed, developed fatty liver (Figure 7B).

All together, these data suggest that, in the liver of females, hepatic ERα contributes to re-arranging the expression of relevant genes, including adropin, to better preserve liver metabolic homeostasis even under an excess of dietary lipids in the liver, thus contributing to tackle fatty liver development in females.

## 4. Discussion

Males and females have a different capability to adapt their metabolism to nutritional cues, including fasting [9,55,56,57], calorie restriction [58,59], protein and amino acids restriction [60,61,62,63], as well as diets enriched in lipids [13,64,65], sugars [66], proteins or specific amino acids [67,68]. The liver plays a key role in the adjustment of systemic metabolism to the challenges mentioned above through extensive regulation of gene expression [69,70,71] and secretion of proteins, including hepatokines, that affect metabolic processes [20]. 

Considering the metabolic relevance of the liver, it is conceivable that female mammals, who are responsible of the metabolic sustainment of the growing fetus, may have developed mechanisms able to propagate hepatic signals and tune other organs’ metabolism accordingly. Given its relevance in hepatic sexual dimorphism under several nutritional conditions [9,13], hepatic ERα orchestrates the metabolic adaptation accordingly to nutrient availability also by regulating hepatokine expression, thus ensuring a tight connection between reproductive and metabolic functions [12,38]. ERα-dependent regulation of hepatokines might thus contribute to the sex-specific susceptibility to cardiometabolic consequences of NAFLD. Of relevance, ERα plays a crucial role in modulating the expression of FGF21 [72], a hepatokine known to have a beneficial effect in limiting NAFLD and NASH [73,74,75]. However, apart from FGF21, sex and estrogen-dependent regulation of hepatokines have been very little investigated.

Our present study identified a subset of four hepatokines, namely *Enho*, *Inhbe*, *Igfbp1*, and *Tsku*, that better display sex and estrogen-dependent regulation in the liver under physiological conditions (Figure 1A–C). Notably, all of them bear classical EREs close to the promoter (Figure 1D,E); furthermore, their mRNA content is modulated by estrogen treatment in HepG2 cells and hepatocytes derived from control but not from LERKO mice, in a sex-specific fashion, confirming the relevance of estrogens and hepatic ERα signaling in their modulation (Figure 2). In this frame, our data go beyond the state of art, since they are the first showing the engagement of hepatic ERα in mediating the sexually dimorphic regulation of these hepatokines in vitro and in vivo, in a physiological context.

In a mouse model of menopause, CE mostly restores the expression of the four hepatokines to the levels of fertile females (Figure 3), suggesting that estrogen-based HRT has the potential to limit the propagation of liver-derived systemic alterations, acting also through the regulation of hepatokines. 

More interestingly, under an excess of dietary lipids, the hepatokines studied show a sex-specific regulation (Figure 4B–E). Among those, adropin is of particular interest because it is the only one whose mRNA content depends on sex and hepatic ERα and negatively correlates with fatty liver in a sex-specific fashion (Figure 4F). These findings agree with other studies reporting that adropin overexpression inhibits lipogenesis, thus contributing to prevent fatty liver [22,23]. In fact, several studies demonstrated that adropin levels are lower in humans with obesity, type 2 diabetes, insulin resistance, hepatic steatosis/NAFLD, polycystic ovary syndrome (PCOS), and cardiovascular diseases [22]. In mice, HFD reduces *Enho* mRNA levels concomitantly with the development of hepatic steatosis, while adropin overexpression attenuates HFD-derived hepatic steatosis, reduces weight gain, increases whole-body fatty acid oxidation, and improves insulin sensitivity [48]. In agreement, hepatic steatosis is exacerbated in adropin-null mice [23]. Acute administration of recombinant adropin lowers the hepatic expression of lipogenesis-associated genes, including *Fasn* [48]. Other studies have shown that high adropin levels improve insulin sensitivity and carbohydrate and lipid metabolism while suppressing hepatic steatosis [23], suggesting that restoring proper adropin levels can be a valuable therapeutic approach for the treatment of NAFLD. 

Our present data are the first to reveal that the physiological induction of adropin mediated by hepatic ERα in the liver of females under HFD negatively correlates with the expression of key genes involved in the synthesis of fatty acids and lipids such as *Acaca*, *Fasn*, *Elovl6*, *Dgat2*, and *Lpin1*, as well as *Pklr*, which has a role in the regulation of DNL (Figure 5 and Figure 6). In addition, the female-dependent induction of adropin under HFD boosts the response to oxidative stress, helping in counteracting the generation of reactive oxygen species (ROS) that promotes NAFLD progression and liver inflammation [32,54].

Notably, fatty acid synthesis is the pathway mostly modified by HFD in the liver of control females (Figure 7B), but not in control males or in LERKO of both sexes, that, on the contrary, are unable to prevent liver lipid deposition. The female-specific and hepatic ERα-dependent reprogramming of liver transcriptome under HFD is sufficient to preserve liver metabolic homeostasis, since very few metabolites show significant changes (Figure 7A). By contrast, the inability of males and of LERKO mice of both sexes to re-arrange the liver transcriptome in response to an excess of dietary lipids results in altered levels of several metabolites (Figure 7A), suggesting that the loss of liver homeostasis can trigger fatty liver and liver degeneration.

Moreover, adropin can influence systemic metabolism by acting on the regulation of substrate utilization in the skeletal muscle [45], thus improving insulin sensitivity and, probably, preventing the development and progression of cardiometabolic diseases. Further research is needed to verify whether the hepatic expression of adropin correlates with adropin circulating levels, and to investigate the extent to which changes in adropin have the potential to prevent or limit the metabolic and endocrine defects associated with fatty liver/NAFLD.

## 5. Conclusions

The present study identified adropin as the hepatokine better predicting sex-specific and hepatic Erα-dependent regulation of hepatic metabolism under different hormonal and nutritional conditions. In particular, the hepatic Erα-mediated induction of adropin under a condition of excess of dietary lipids limits lipogenesis and lipid deposition in the liver of females, suggesting that the restoration of adropin levels represents a valuable sex-specific therapeutic approach for the treatment of NAFLD and, possibly, of related cardiometabolic consequences.

## Figures and Tables

**Figure 1 nutrients-14-03262-f001:**
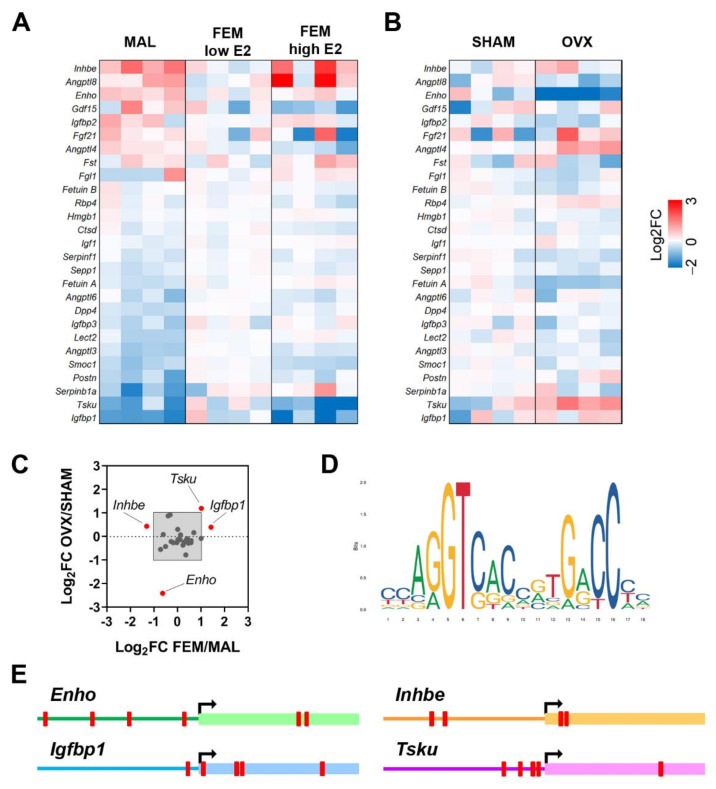
Sex and estrogen levels affect hepatokine mRNA levels in the liver of mice under physiological conditions. (**A**,**B**) Heatmaps reporting as Log_2_ fold change (FC) the mRNA levels of the most relevant hepatokines known to be involved in NAFLD from RNA-Seq analysis performed in the livers of males (MAL) and females (FEM) at low and high levels of estrogens (**A**) and in the livers of sham-operated (SHAM) and ovariectomized (OVX) female mice (**B**) (*n* = 4). (**C**) Cluster analysis performed by plotting the Log_2_ FC of fertile females (FEM) over males (MAL) against the Log_2_ FC of OVX over SHAM females. *Enho,* adropin; *Inhbe*, inhibin subunit beta E; *Igfbp1*, insulin like growth factor binding protein 1; *Tsku,* tsukushi. (**D**,**E**) Estrogen-responsive elements (EREs) identified in the region −/+1000 around TSS of *Enho*, *Inhbe*, *Igfbp1*, and *Tsku* (EREs are shown as red blocks).

**Figure 2 nutrients-14-03262-f002:**
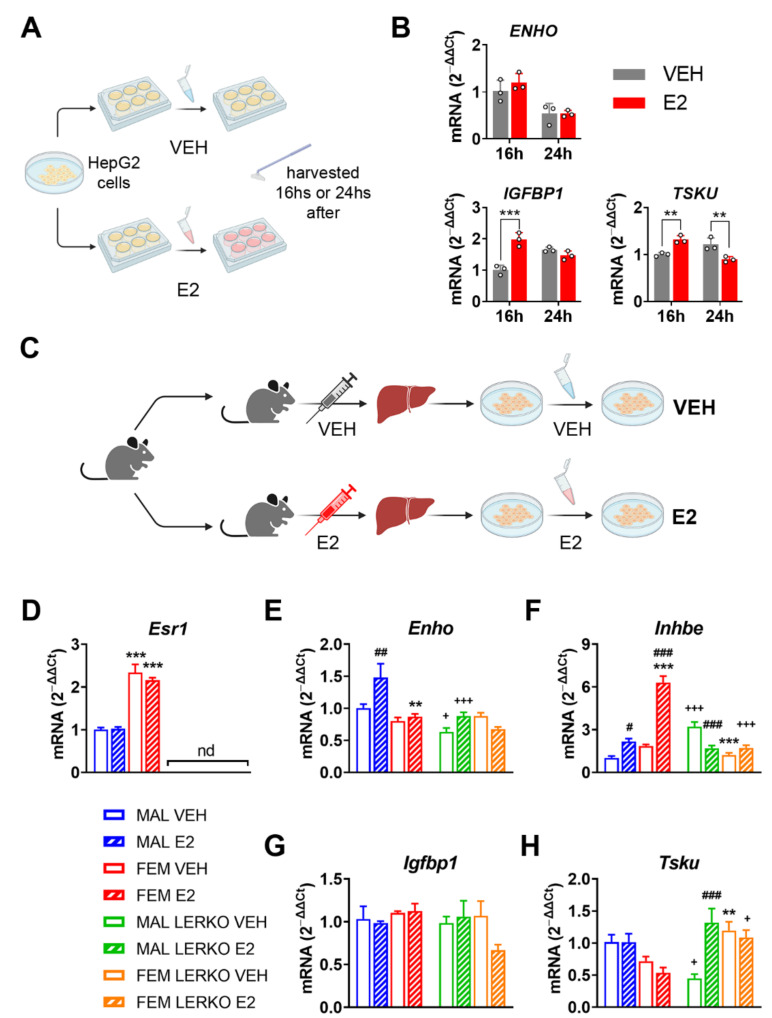
Estrogen modulates hepatokine mRNA levels in hepatic cells acting through ERα. (**A**) Experimental design employed to evaluate the effects of estrogen treatment in HepG2 cells. (**B**) mRNA content of *ENHO*, *IGFBP1*, and *TSKU* measured by RT-PCR in HepG2 cells treated with vehicle (VEH) or 1 nM 17β-estradiol (E2) for 16 or 24 h. Data are presented as mean ± SEM (*n* = 3). ** *p* < 0.01 and *** *p* < 0.001 vs. VEH-treated cells. (**C**) Experimental design employed to evaluate the effects of estrogen treatment in primary hepatocyte cells derived from male and female control or LERKO mice treated with VEH or E2. (**D**–**H**) mRNA content of *Esr1* (**D**), *Enho* (**E**), *Inhbe* (**F**), *Igfbp1* (**G**), and *Tsku* (**H**) measured by RT-PCR in hepatocyte cells derived from male and female control or LERKO mice treated with VEH or E2. nd = not determined. Data are presented as mean ± SEM (*n* = 3–4). ** *p* < 0.01 and *** *p* < 0.001 vs. male-derived hepatocytes; # *p* < 0.05, ## *p* < 0.01 and ### *p* < 0.001 vs. VEH-treated cells; + *p* < 0.05 and +++ *p* < 0.001 vs. control-derived hepatocytes by two-way ANOVA followed by Bonferroni’s *post hoc* test.

**Figure 3 nutrients-14-03262-f003:**
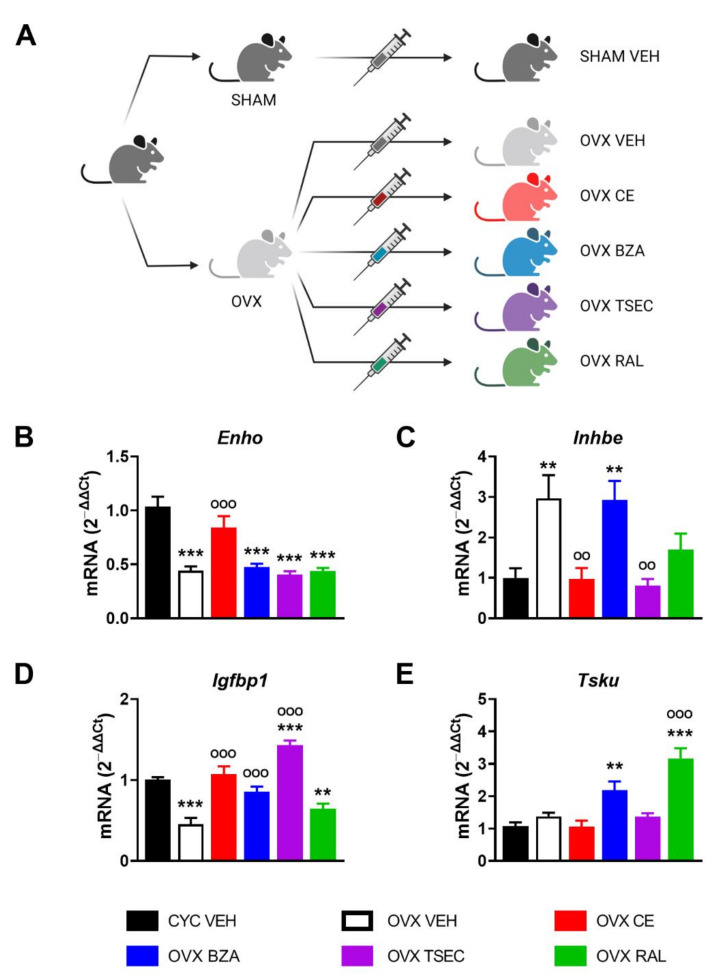
Estrogen-based hormone replacement therapy (HRT) modulates hepatokine mRNA levels in ovariectomized (OVX) female mice. (**A**) Experimental design adopted to evaluate the effects of several HRTs in OVX female mice. (**B**–**E**) mRNA content of *Enho* (**B**), *Inhbe* (**C**), *Igfbp1* (**D**), and *Tsku* (**E**) measured by RT-PCR in the liver of cycling fertile (CYC) and OVX female mice treated with vehicle (VEH), conjugated estrogen (CE), bazedoxifen (BZA), BZA *plus* CE (TSEC), or raloxifen (RAL) for 21 days. Data are presented as mean ± SEM (*n* = 8–10). ** *p* < 0.01 and *** *p* < 0.001 vs. CYC; °° *p* < 0.01 and °°° *p* < 0.001 vs. OVX by one-way ANOVA followed by Bonferroni’s *post hoc* test.

**Figure 4 nutrients-14-03262-f004:**
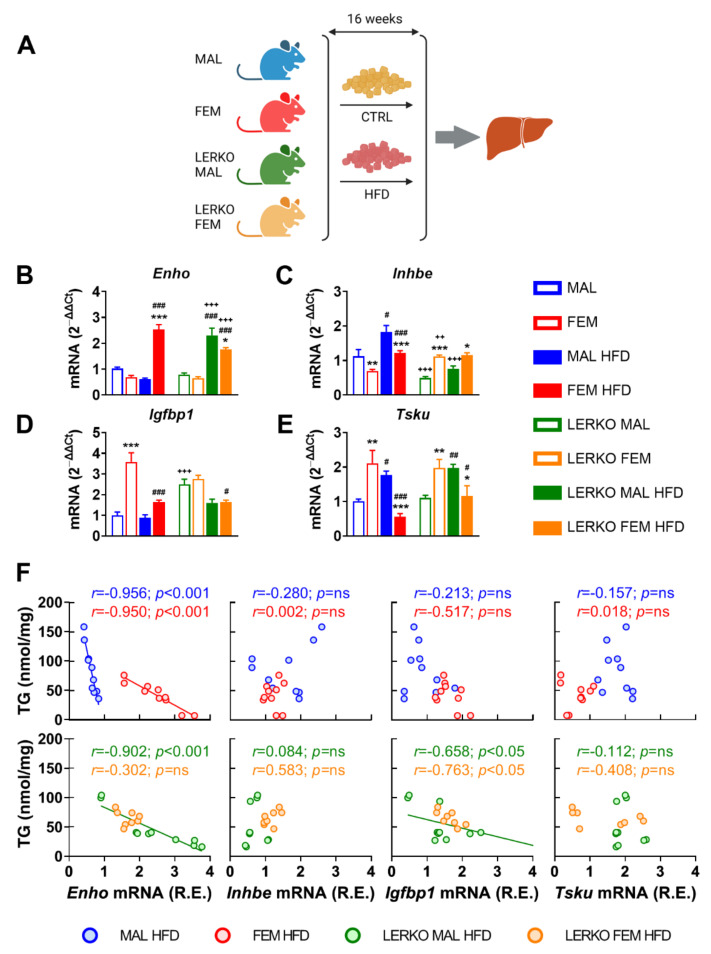
*Enho* mRNA levels negatively correlate with fatty liver in a HFD-induced mouse model of NAFLD. (**A**) Experimental design adopted to evaluate the relevance of estrogen signaling in the liver metabolic response to a high-fat diet (HFD). (**B**–**E**) mRNA contents of *Enho* (**B**), *Inhbe* (**C**), *Igfbp1* (**D**), and *Tsku* (**E**) measured by RT-PCR in the liver of male and female control and LERKO mice fed with control diet or HFD. Data are presented as mean ± SEM (*n* = 8–10). * *p* < 0.05, ** *p* < 0.01, and *** *p* < 0.001 females vs. males; # *p* < 0.05, ## *p* < 0.01, and ### *p* < 0.001 HFD vs. control diet; ++ *p* < 0.01 and +++ *p* < 0.001 LERKO vs. control mice by two-way ANOVA followed by Bonferroni’s *post hoc* test. (**F**) Pearson’s correlation and linear regression between liver triglyceride (TG) content and relative expression (R.E.) of *Enho*, *Inhbe*, *Igfbp1*, and *Tsku* mRNA levels measured in the liver of male and female control and LERKO mice fed with HFD.

**Figure 5 nutrients-14-03262-f005:**
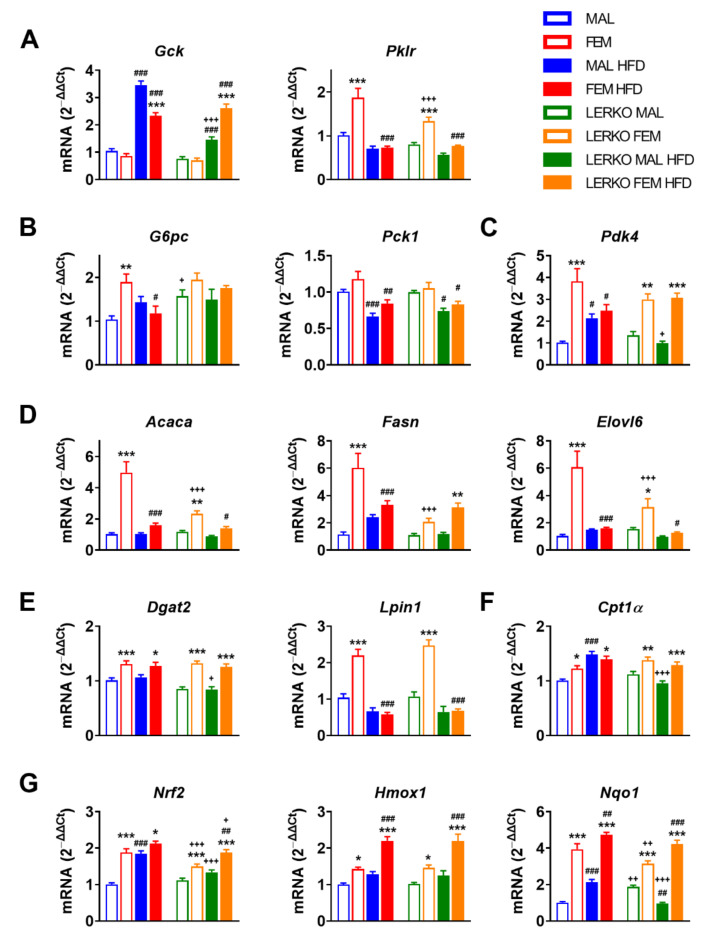
Sex-specific metabolic adaptation to HFD depends on hepatic ERα. (**A**–**G**) mRNA contents of key genes involved in glycolysis (**A**), gluconeogenesis (**B**), regulation of glucose metabolism (**C**), fatty acid synthesis and elongation (**D**), triglyceride synthesis (**E**), fatty acid catabolism (**F**), and oxidative stress response (**G**) measured by RT-PCR in the liver of male and female control and LERKO mice fed with control or HFD. Data are presented as mean ± SEM (*n* = 8–10). * *p* < 0.05, ** *p* < 0.01 and *** *p* < 0.001 females vs. males; # *p* < 0.05, ## *p* < 0.01 and ### *p* < 0.001 HFD vs. control diet; +*p* < 0.05, ++ *p* < 0.01 and +++ *p* < 0.001 LERKO vs. control mice by two-way ANOVA followed by Bonferroni’s *post hoc* test. *Gck*, glucokinase; *Pklr*, pyruvate kinase*; G6pc*, glucose-6-phosphatase*; Pck1*, phosphoenolpyruvate carboxykinase 1*; Pdk4*, pyruvate dehydrogenase kinase 4*; Acaca*, acetyl-CoA carboxylase alpha*; Fasn*, fatty acid synthase*; Elovl6*, ELOVL fatty acid elongase 6*; Dgat2*, diacylglycerol O-acyltransferase 2*; Lpin1*, Lipin 1*; Cpt1a*, carnitine palmitoyltransferase 1A*; Nrf2*, nuclear factor erythroid 2–related factor 2*; Hmox1*, heme oxygenase 1*; Nqo1*, NAD(*p*)H quinone dehydrogenase 1.

**Figure 6 nutrients-14-03262-f006:**
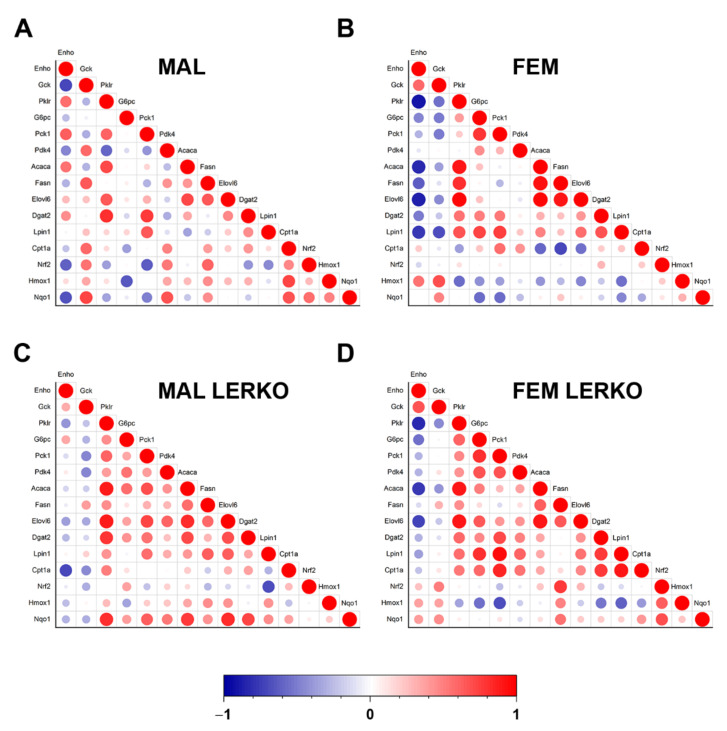
Adropin mRNA content negatively correlates with lipid synthesis in the liver of female mice. (**A**–**D**) Correlation scatter plots showing the positive (in red) and negative (in blue) correlation between mRNA levels of *Enho* and key genes involved in glycolysis, gluconeogenesis, regulation of glucose metabolism, fatty acid synthesis and elongation, triglyceride synthesis, fatty acid catabolism, and oxidative stress response measured by RT-PCR in the liver of control males (**A**), control females (**B**), LERKO males (**C**), and LERKO females (**D**). The color-coded scale indicates the correlation coefficient values.

**Figure 7 nutrients-14-03262-f007:**
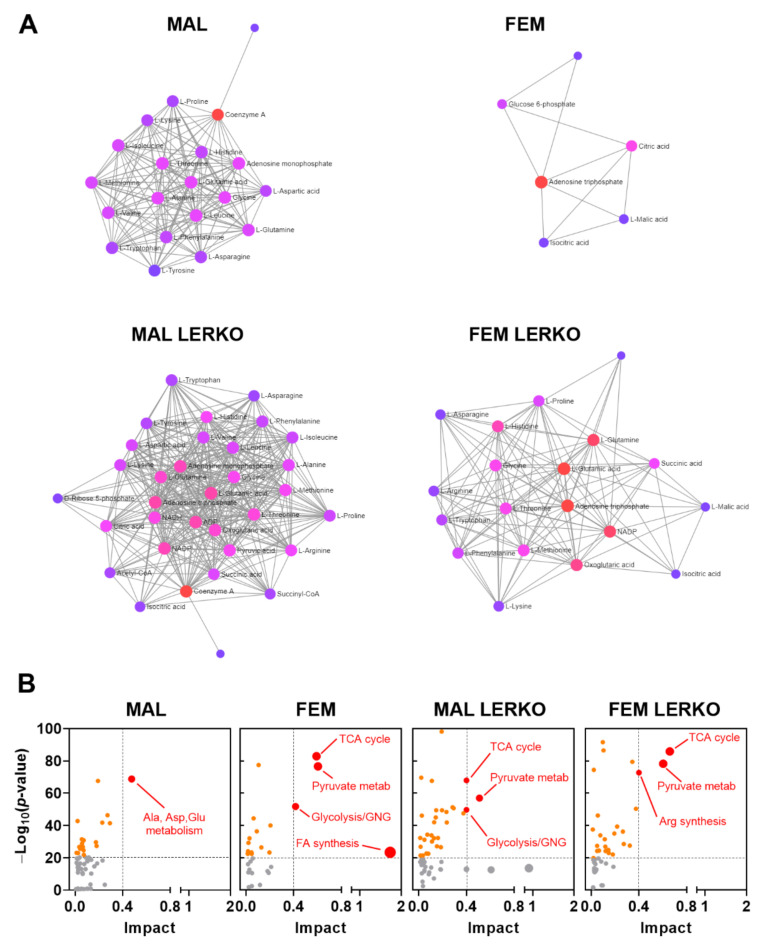
Hepatic ERα contributes to preserve metabolic homeostasis in the liver of females under an excess of dietary lipids. (**A**) HFD effects on metabolite–metabolite interaction in the liver of control males, control females, LERKO males, and LERKO females. (**B**) Metabolic impact analysis of the most relevant biological pathways modified by HFD in the livers of control males, control females, LERKO males, and LERKO females identified through integrative analysis of the changes in the expression of key genes and metabolites. The metabolic pathways are represented as circles according to their enrichment (*p*-value, Y axis) and topology analyses (pathway impact, X axis) using MetaboAnalyst 5.0 software. The color of each metabolic pathway is related to the *p*-value obtained from enrichment analysis and its size represents the fold enrichment score; darker circle colors indicate more significant changes of metabolites in the corresponding pathway. The size of the circle corresponds to the pathway impact score and is correlated with the centrality of the involved metabolites.

## Data Availability

Raw RNA-Seq data have been deposited to Bioproject with the following accession numbers: PRJNA395963 and PRJNA778593.

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
