# Peer review of "ERα-Dependent Regulation of Adropin Predicts Sex Differences in Liver Homeostasis during High-Fat Diet"

_nutrients, 2022, doi:10.3390/nu14163262_

Round 1

Reviewer 1 Report

In this manuscript, Della Torre and coworkers studied the expression of hepatokine genes correlate with physical factors, including sex, liver estrogen receptor alpha level, hormone treatment, and excess dietary lipids. The expressions of a set of genes downstream of the ERalpha were systematically evaluated. One of the hepatokine, adropin, negatively correlates with lipid deposition under high-fat diet in female, showing potential in treatment of NAFLD. The experimental design in manuscript is straightforward and well documented. The quality of this manuscript is suitable for publication on Nutrients. Only few minor expression needs to be corrected.

The authors mentioned "hepatokine expression" or "expression of hepatokines" many times in the manuscript. Given that hepatokines are proteins synthesized and secreted by liver, these descriptions are easily understood as protein level expression of hepatokine genes. However, the authors only focused on the mRNA expression of hepatokine genes. It will be better to use "RNA-level expression of hepatokine genes" to avoid confusion to readers.

In method 2.4, "reverse transcription quantitative (real-time) PCR" is the method that the authors used to evaluate mRNA levels. 

Author Response

The authors mentioned "hepatokine expression" or "expression of hepatokines" many times in the manuscript. Given that hepatokines are proteins synthesized and secreted by liver, these descriptions are easily understood as protein level expression of hepatokine genes. However, the authors only focused on the mRNA expression of hepatokine genes. It will be better to use "RNA-level expression of hepatokine genes" to avoid confusion to readers.

Reply: throughout the study, “hepatokine expression” refers to the mRNA level, as correctly highlighted by this Reviewer. To avoid any confusion or misunderstanding, we changed the text so that the reader can easily understand that we are referring to mRNA levels of these hepatokines.

Moreover, nomenclature (i.e. Enho when referring to adropin mRNA levels in mice) should help the reader in understanding that we are referring to mRNA levels of genes.

In method 2.4, "reverse transcription quantitative (real-time) PCR" is the method that the authors used to evaluate mRNA levels.

Reply: changed (see page 4, line 149and line 166).

Reviewer 2 Report

NAFLD is an important medical problem, and the keys to understanding many mechanisms of NAFLD pathogenesis are still not available to clinicians. This article analyzes the role of hepatokines in the sex-specific development of NAFLD using modern methods.

Comments:

1.   It is recommended not to use abbreviations (HFD) in the title of the article. It is recommended to write all abbreviations in full when first mentioned.

2. It would be useful to add a scheme with the study design in the materials and methods.  It is also recommended that the materials and methods state how many mice were in each group. Were hormone levels in the mice controlled in the experiment?  For example, to exclude the effects of adrenal estrogens.

3.   It is recommended to add information on Bioproject PRJNA395963 and PRJNA778593, which are analyzed in section 3.1, to the materials and methods section.

Author Response

Comments:

  1. It is recommended not to use abbreviations (HFD) in the title of the article. It is recommended to write all abbreviations in full when first mentioned.

Reply: we modified the title as requested by the Reviewer (see page 1, line 3). We checked the text for abbreviations and we modified it to be sure to write all abbreviations in full when first mentioned.

  1. It would be useful to add a scheme with the study design in the materials and methods. It is also recommended that the materials and methods state how many mice were in each group. Were hormone levels in the mice controlled in the experiment? For example, to exclude the effects of adrenal estrogens.

Reply: the authors think it would be redundant to add other study design in the Materials & Methods section, since the schemes of the study design are already depicted in the Figures (see Figures 2A, 2C, 3A, and 4A).

The number of mice in each experimental group is made explicit in the related legend Figure (i.e.: In the Figure 2b: …Data are presented as mean ± SEM (n=3). In the Figure 2C: Data are presented as mean ± SEM (n=3-4). And so on..).

The phase of estrous cycle of females is assessed by vaginal smears, a non-invasive methodology that allows to determine the phase of the estrous cycle very accurately with very low distress for mice. Alternatively, we should measure the levels of sex hormones: this methodology, however, requires a very huge amount of blood (>250 ul) and the use of very sensitive methods and instrumentation (MS/MS), that, however, can not ensure to discriminate among phases with very low levels of estrogens such as estrus, metestrus and diestrus. In our experience, vaginal smears analysis is the best and more reliable method to assess the phase of the estrous cycle in females. 

Despite the MS/MS analysis sensitivity, the assessment of sex hormones, especially estrogen levels, in mice is still a challenge; in several cases (i.e. ovariectomy) the levels of estrogens measured are below the limit of detection. Also for this reason, we believe that effects of adrenal estrogens can be minimal with respect to ovarian-derived estrogens.

  1. It is recommended to add information on Bioproject PRJNA395963 and PRJNA778593, which are analyzed in section 3.1, to the materials and methods section.

Reply: information on Bioproject PRJNA395963 and PRJNA778593 is already made explicit in the Materials & Methods section (see page 3, lines 119-148). However, since the Editor asked us to avoid repetition, we have removed this part of the text and we now refer to our previously published studies that report all the related information about these 2 Bioprojects.